# Can Our Blood Help Ensure Antimicrobial and Anti-Inflammatory Properties in Oral and Maxillofacial Surgery?

**DOI:** 10.3390/ijms24021073

**Published:** 2023-01-05

**Authors:** Lana Micko, Ilze Salma, Ingus Skadins, Karina Egle, Girts Salms, Arita Dubnika

**Affiliations:** 1Institute of Stomatology, Riga Stradins University, LV-1007 Riga, Latvia; 2Department of Oral and Maxillofacial Surgery, Riga Stradins University, LV-1007 Riga, Latvia; 3Baltic Biomaterials Centre of Excellence, Headquarters at Riga Technical University, LV-1658 Riga, Latvia; 4Department of Biology and Microbiology, Riga Stradins University, LV-1007 Riga, Latvia; 5Rudolfs Cimdins Riga Biomaterials Innovations and Development Centre of RTU, Faculty of Materials Science and Applied Chemistry, Institute of General Chemical Engineering, Riga Technical University, LV-1007 Riga, Latvia

**Keywords:** oral and maxillofacial surgery, autologous materials, biomaterials, platelet concentrate, platelet-rich fibrin, antimicrobial, anti-inflammatory, platelet-rich fibrin properties

## Abstract

In recent decades, the potential of PRF has been extensively studied. The number of studies about PRF has increased three times since the year 2012, but the full spectrum of its fundamental properties, such as antimicrobial and anti-inflammatory activity, is not clearly described. In oral and maxillofacial surgery, PRF is described in alveolar ridge preservation, orthognathic surgery, cleft lip and palate surgery, maxillary sinus augmentation, and dental implant placement as demonstrating favorable results and its clinical advantages. The structural complexity, inhomogeneous nature, and clotting ability of PRF make its antimicrobial effect evaluation complicated. Nevertheless, most of the used antimicrobial testing methods are based on antibacterial agent diffusion ability in culture media. Because the oral and maxillofacial region is the most frequent area of PRF application, its antimicrobial activity evaluation also prevails in the oral microbiome. PRF’s biological potential is highly dependent on the specific preparation protocol and methodology used; it should be carefully prepared and kept under proper conditions to keep cellular content alive. PRF’s influence on living cells demonstrates a stimulating effect on bone regeneration, and an angiogenetic effect, and it provides anti-inflammatory activity. According to analyzed studies, PRF demonstrated success in oral and maxillofacial surgery in various methods of application. Antibacterial and anti-inflammatory properties were proven by antibacterial activity against different bacterial species, sustained growth factor, sustained release, and cell activity on the material application. Accurately and correctly prepared PRF can ensure antibacterial and anti-inflammatory properties, and it can be a beneficial clinical tool in oral and maxillofacial surgery.

## 1. Introduction

Over the years, modern medicine has expanded, and regenerative medicine as a subfield is one of the emerging and driving forces for this growth. Body tissue has a highly organized structure and unique composition; therefore, for the successful replacement of such a complex structure, the implant material should not only promote a regenerative process, but also possess certain mechanical properties and create a perfect microenvironment for cell growth and differentiation [1,2]. Autologous materials, such as platelet-rich fibrin (PRF), can fulfill the criteria of being a biodegradable biopolymer by creating a three-dimensional structure with enclosed platelet-derived growth factors, thus facilitating cell adhesion and proliferation [3].

This review provides insight into a wide range of clinical applications of PRF in oral and maxillofacial surgery (alveolar ridge preservation, orthognathic surgery, cleft lip and palate surgery, sinus lift, and dental implant placement) and analysis of PRF antimicrobial and anti-inflammatory properties from in vitro studies. We aim to emphasize the potential of PRF use in medicine by describing its basic antimicrobial and immunological properties as well as to demonstrate the reasons for the controversial evaluation of PRF’s effects.

This review focuses on PRF, which is a fully autologous biomaterial prepared from a patient’s own blood by using collection tubes and only one centrifugation process. Materials that have additives or additional biomechanical processing were excluded from this review. Such additives as anticoagulants (citrate phosphate dextrose adenine (CPDA), sodium citrate), coagulation activators (calcium glutamate or bovine thrombin, calcium chloride), and gel agents may cause changes in the molecular and mechanical properties of platelet concentrates (PCs). As described by Giannini et al., the use of an anticoagulant, a gel agent, and a coagulation activator in the preparation of plasma rich in growth factors (PRGF) can influence the thickening of the fibrin fibers by forming a rigid mesh with improved mechanical properties but impaired cellular migration and cytokine release, thus influencing antimicrobial and immunological properties [4,5]. Furthermore, sodium citrate itself provides antimicrobial activity [6]. Thereby, the addition of such materials would cause nonphysiological changes in the biomaterial.

Despite the complexity of mimicking parts of the body, tissue regeneration may offer a novel treatment for injuries, various soft and hard tissue defects, malfunctioning, and aging. One of the major concerns of the use of biomaterials is the risk of potential tissue infections [7]. Infections of such foreign material can be severe and difficult to treat [8]. Whitman et al. were the first to describe the use of PCs, and further studies made PCs a promising biomaterial for various medical fields [9,10]. PCs are whole blood products characterized by a supraphysiological concentration of platelets and growth factors [11]. To replace the use of fibrin glue, two main types of PCs were introduced: platelet-rich plasma (PRP) and platelet-rich fibrin (PRF). A wide range of different preparation protocols was analyzed by Dohan et al. [10].

In general, the classification of PCs is divided into four main families based on cell content and fibrin density (Table 1) [12,13].

PRP is obtained in several steps and done in various ways. The first centrifugation step separates red blood cells (RBCs), platelet-poor plasma (PPP), and platelet-rich plasma (PRP) or buffy coat. The process is followed by PRP collection and application using different additives, such as thrombin, calcium chloride, or others, to initiate platelet activation and fibrin polymerization [10]. A great number of various protocols allow collection of P-PRP, also called leukocyte-poor plasma, using automated protocols (plasmapheresis with a laboratory cell separator and Vivostat PRF) and manual protocols (modified P-PRP: Anitua’s PRGF). Likewise, leucocyte and platelet-rich plasma (L-PRP) can be prepared using various automated protocols (SmartPReP, PCCS, GPS, and Magellan) and manual protocols, respectively (Curasan, Friadent-Schutze, Regen, and Plateltex) [10].

Using the term PRF, two PC products can be described. P-PRF, also called leucocyte-poor platelet-rich fibrin, is obtained using two centrifugation steps. The first centrifugation is just after mixing blood with tri-sodium citrate and a separator gel, which is followed by a second centrifugation to get platelet-rich fibrin matrix (PRFM) clot from PPP, buffy coat, and CaCl_2_, thus triggering the coagulation process [10]. The previously mentioned PRF-obtaining protocol cannot be considered “natural”, thus there are also other PRF preparing methods with which PRF is made with no biochemical processing from nonanticoagulated blood. Venous blood is collected in glass tubes or silicon-coated plastic tubes without any additives and immediately centrifugated. After the natural coagulation process, three different layers are observed: the red blood cell base layer, the PRF clot or injectable PRF, and the acellular plasma top layer [10,14]. In this huge diversity of PCs, such PRF differs from others by the simplicity of the technique as well as its lower cost, purity, and product quality. Such PRF is also highlighted among other biomaterials with its improved biomechanical properties due to the higher density of the extracellular matrix and its architecture [10]. According to the further introduced “low-speed concept”, Choukroun classified other PRF preparations shown in Table 2. Lower-speed centrifugation allows obtaining injectable PRF with a higher cell count (platelets, leukocytes) and a greater amount of growth factors [15].

For these reasons, purely autologous PRF without any additives is a superior surgical additive for tissue regeneration, and it has shown great potential to be used in regenerative medicine [17]. PRF is extensively studied in various fields and has reached acceptance and practical applications in medicine, including being used in maxillofacial, plastic, and reconstructive surgery [18,19].

Despite the fact that PCs have been known for several decades and are widely described, there are controversial results about their clinical efficacy and a lack of structured and complete knowledge about their properties. Among other properties, such basic ones as antimicrobial and anti-inflammatory action may have a significant role in the proper and successful application of a biomaterial for tissue healing and regeneration. Any surgical manipulation in the oral and maxillofacial region is subjected to the most diverse spectrum of microorganisms [20]. Bacteria found in specific areas can cause severe diseases as well as surgical site infections. For example, infectious complications after sinus lift surgery are found to occur after around 12.5% [21] of surgeries, and the rate of surgical site infection (SSI) in orthognathic surgery is reported to be from 1.4% to 33.4% [22]. Antibiotic prophylaxis is used when the risk of SSI is increased [23]. Thereby, a biomaterial with antimicrobial and anti-inflammatory action can reduce the need for antibiotics, improve tissue healing, and may be the superior treatment option for infectious complications. 

## 2. PRF Clinical Application in Oral and Maxillofacial Surgery

In the last decade, between 2012 and 2022, the greatest number of studies on PRF clinical applications published in the PubMed database were done in the field of oral and maxillofacial surgery, dealing with various diseases, injuries, and defects in the oral cavity, jaws, face, and neck. PRF use was described in alveolar ridge preservation [24,25,26,27,28,29,30,31,32], orthognathic surgery [33,34], cleft lip and palate surgery [35,36], maxillary sinus augmentation [37,38,39,40,41,42,43,44], and dental implant placement [24,45,46,47,48,49]. It showed the ability to accelerate the closure of an extraction socket, improve periodontal status, reduce postoperative pain sensation and swelling, accelerate neurosensory recovery, reduce bone resorption, increase and accelerate bone formation, enhance bone stability after surgery, and provide faster dental implant osseointegration as well as increased dental implant stability (Figure 1). The majority of studies confirm the clinical advantages of PRF; thus, a deeper understanding of its use in several applications is a must. 

### 2.1. Alveolar Ridge Preservation

One of the most common surgical manipulations in dentistry and maxillofacial surgery is dental extraction, which leads to physiological changes in the oral cavity. After tooth extraction, the alveolar process of the maxilla and mandible, which is a tooth-dependent structure, undergoes bone resorption in the vertical and horizontal dimensions [50]. In general, bone resorption occurs due to disuse atrophy, lack of blood supply, and inflammation, and it is the result of local and systemic factors, such as periapical infection, chronic periodontitis, loss of socket wall integrity, surgical trauma, smoking, genetic disorders, and immunosuppression. Alveolar ridge preservation, a method of decreasing bone resorption after tooth extraction for future prosthodontic treatment, involves the use of various biomaterials [51]. Currently, it is known that several grafting materials may possess some disadvantages, as they can remain in the extraction socket for a longer time than six months and possibly decrease bone density as well as decrease implant–bone contact after implant placement [51,52]. A biomaterial with a specific resorption rate could promote bone formation, regulate angiogenesis and inflammation, and would be advantageous to fill the extraction socket. Previous studies describe the use of PCs that could be the source of autologous growth factors for alveolar ridge preservation after extraction. 

During the last 10 years, several studies showed lower bone resorption after tooth extraction by applying PCs. Canellas et al., used L-PRF (centrifugated at 2700 rpm, 408× *g*, 12 min) to fill nonmolar tooth extraction sockets. Computed tomography three months after extraction revealed significantly lower bone resorption 1 mm below the crest: 0.93 ± 0.9 mm for the L-PRF group and 2.27 ± 1.2 mm for the control group (*p* = 0.0001) [25]. Similarly, Temmerman et al., after symmetrical tooth extractions, filled sockets at one side with L-PRF (2700 rpm, 408× *g*, 12 min) and covered them with L-PRF membranes, whereas extraction sockets on the other side were treated as control sites. Three months after the tooth extractions, cone beam computed tomography (CBCT) was done and analyzed, concluding that the use of L-PRF was beneficial because it resulted in a significant reduction of bone width resorption −1 mm below alveolar crest level (22.84%) in comparison to control sites (51.92%). In addition, bone filling in the sockets was significantly different between test (94.7%) and control (63.3%) extraction sockets [50]. Different results were shown by Castro et al.: no efficacy of L-PRF (centrifugated using Intra-Spin, Intra-Lock system, at 2700 rpm, 408× *g*, 12 min) and A-PRF+ (Duo Process, 1300 rpm, 145× *g*, 8 min) was demonstrated, as the reduction of vertical and horizontal dimension changes showed similar results at −1 mm below the crest in all sockets (*p* > 0.05). In this study, at least three neighbor teeth were extracted, leaving one socket without PC filling, and the CBCT evaluation was done three months after the procedure. However, the percentage of socket fill as well as the percentage of bone and tissue volume when L-PRF and A-PRF+ were used were both found to be statistically significantly higher (*p* < 0.05) and showed an accelerative effect of PCs on bone formation [26]. 

In addition, Suttapreyasri et al., did not show a statistically significant decrease in bone resorption using A-PRF (3000 rpm, 10 min) after premolar extractions compared to the natural healing [53]. Hauser et al., demonstrated that not just the filling material is important, but also the extraction complexity. The efficiency of an L-PRF (2700 rpm, 12 min) membrane was shown in the quality of bone harvested during implant placement 8 weeks after extraction. Bone tissue qualities after healing of simple extraction sockets (filled with L-PRF), extraction sockets with mucoperiosteal flaps (both filled with PRF membranes), and simple extraction sockets (without filling) were compared. Bone samples showed statistically significantly higher elastic moduli (+17.42%) and working energy (+11.05%) in sockets that underwent extraction without mucoperiosteal flaps and using PRF as filling material. Furthermore, microcomputed tomography in PRF-filled sockets showed an increased trabecular number (+28.1% and +22.2%) and lower trabecular separation (−35.3% and −26.6%) compared to the control group and to sockets filled with PRF after complicated extractions [27]. Clark et al., compared the efficiency of A-PRF (1300 rpm, 200× *g*, 8 min), freeze-dried bone allograft (FDBA), and their combination with that of blood clot for bone formation after tooth extraction. Measures done 15 weeks after extraction revealed significantly less ridge height reduction using A-PRF (1.8 ± 2.1) or A-PRF with FDBA (1.0 ± 2.3) compared to the blood clot group (3.8 ± 2.0) [28].

Histomorphometric analysis done by Areewong et al., two months after L-PRF (centrifugated at 2700 rpm, 408× *g*, 12 min) was used as filling for single-rooted tooth extraction sockets also showed a higher bone formation ratio in the L-PRF group (31.33 ± 18%) in comparison to naturally healed sockets (26.33 ± 19.63%) [29]. The histological and radiological analysis in a clinical study by Castro et al. reported more newly formed bone for the PRF group, but the results were not statistically significant [26]. Martins et al., compared maxillary frontal tooth extraction sockets filled with PRF (400× *g*, 12 min) not only to naturally healed sockets, but also to sockets filled with bone marrow aspirate concentrate (BMAC) and PRF. Six months after the operation, bone core was harvested and histomorphometric analysis was done that showed a significantly higher amount of mineralized tissue in sockets filled with PRF (54.2 ± 4.31%) as well as with PRF–BMAC (64.7 ± 6.74%) compared to naturally healed sockets (41.6 ± 5.98%). The difference between PRF and PRF–BMAC groups was 10%, thus indicating the trend of more bone formation using PRF in combination with BMAC [30]. Microcomputed tomography done in another study by Clark et al., demonstrated significantly more vital bone 15 weeks after extraction for the extraction sockets filled with A-PRF (1300 rpm, 200× *g*, 8 min) (46% ± 18%) in comparison to sockets where FDBA (29% ± 14%) was used [28].

In various studies, PRF was studied for pain reduction after extraction. Kumar et al., did not get statistically significant results; however, the study demonstrated no pain or swelling for 24 h after tooth extractions when PRF (3000 rpm, 10 min) was used for filling [31]. Similarly, Temmerman et al. noted less postoperative pain and discomfort, especially during the early phases of healing at days three, four, and five, in patients who received treatment with L-PRF (2700 rpm, 408× *g*, 12 min) filling and a membrane after symmetrical tooth extractions in comparison to the unfilled control socket [50].

Marenzi et al., described the effect of L-PRF (2700 rpm, 408× *g*, 12 min) for use in tooth extraction sockets. It was shown that by using L-PRF, a significant reduction in pain could be achieved, as patients reported a lower postextraction pain mean value (3.2 ± 0.3) in comparison to extraction without PCs (4.5 ± 0.7). Statistically significantly faster socket closure was achieved using treatment with L-PRF and, at day seven, the modified healing indices were 4.8 ± 0.6 and 5.1 ± 0.9 for the experimental group sockets and test group sockets, respectively [32].

### 2.2. Orthognathic Surgery

Orthognathic surgery is done to correct dentofacial deformities to restore anatomical and functional relations. Le Fort I osteotomy and bilateral sagittal split osteotomy of the mandibular ramus are done for repositioning the maxilla and mandibula and redefining the face [54].

Despite rigid fixation of the bones using titanium miniplates and screws, bony tissue relapse may occur in the postoperative period, especially in the long term. If the size of the gap between the osteotomy segments is larger than 3 mm, then bone healing tends to be inadequate, which may result in impaired hard tissue formation that replaces it with fibrotic tissue [33]. A higher relapse rate is reported in unilateral cleft lip and palate patients and those who underwent bimaxillary surgery [55]. In such cases, the use of bone grafts should be considered [33]. Another complication that can occur after orthognathic surgery is a neurosensory disturbance (NSD). While several factors contribute to NSD after bilateral sagittal split osteotomy, such as inferior alveolar nerve position, mandibular movement magnitude, etc., complications occur in up to 84.6% of patients [34]. Despite the fact that PRF is rarely described as a biomaterial for use in orthognathic surgery, in some research it has demonstrated favorable effects [34].

In the study done by Tabrizi et al., PRF (2800 rpm, 12 min) was applied in 21 patients during bilateral sagittal split osteotomy in the mandibula to one side before fixation, leaving the other side as the control to evaluate neurosensory recovery. The neurosensory evaluation was done six and twelve months after surgery using a two-point discrimination (TPD) test (i.e., showing the minimal recognizable distance between two pinpricks), a brush directional stroke test (i.e., determining the true direction of a brush stroke on the chin), and a self-report of paresthesia using a 10-point visual analogue scale (VAS). Using the TPD test, it was found that six and twelve months after surgery, the minimal recognizable distance between two pinpricks was smaller with a statistical significance (*p* = 0.001) at the sides where PRF was used at both time points. The brush directional test done six and twelve months after surgery also showed a statistically significant difference (*p* = 0.001) between both sides as more patients reported the correct directions in the test side. Statistically significant were also differences regarding self-reported paresthesia (*p* = 0.001); the mean self-reported paresthesia VAS scores six and twelve months after surgery were higher in the test side where PRF was used to fill the osteotomy gap [34]. In another study, Tabrizi et al. described the use of PRF in patients who underwent Le Fort I osteotomy for maxillary advancement, where PRF (2800 rpm, 14 min) was placed in osteotomy sites after fixation for patients in the test group. For others, no biomaterial was used to fill the gap. After 12 months, a comparison of lateral cephalograms (immediately after surgery and after 12 months) was done to determine the amount of maxillary relapse at point A in relation to the x- and y-axis. Results showed a relapse in the test group of 0.45 ± 0.67 mm and 0.77 ± 1.15 mm to the x- and y-axis, respectively, which was significantly smaller than in the control group patients results of 1.86 ± 0.56 mm and 2.25 ± 1.22 mm to the x- and y-axis, respectively. The mean relapses to the x- and y-axis showed a statistically significant difference between both groups (*p* < 0.001). With this research, authors highlighted the PRF’s ability to enhance the stability of the maxilla after Le Fort I osteotomy and recommended the use of PRF in this type of surgery whenever possible [33].

### 2.3. Cleft Lip and Palate Surgery

The alveolar cleft is a bony defect in the alveolar process and affects 75% of cleft lip or cleft lip and palate patients [56]. Without reconstruction or even failed reconstruction, it may result in the presence of an oronasal fistula, fluid reflux, speech pathology, growth deficiency of the maxilla, teeth eruption problems, lack of bone support for anterior teeth, dental crowding, and effects on facial growth [35,56]. Alveolar cleft reconstruction using an autogenous bone graft from the ilium is considered the gold standard [56]. However, there is great interest in bone grafting additives, such as PCs, to promote and enhance bone graft healing [35], and some studies have already described positive effects of using autogenous bone in combination with differently prepared PRF. 

Dayashankara Rao et al., used both i-PRF (700 rpm, 3 min) and A-PRF membranes (1300 rpm, 8 min) together with an iliac bone graft for either secondary alveolar bone grafting or bone grafting alone in the cleft alveolus in unilateral cleft lip and palate patients aged seven years and older. Three and six months postoperatively, intraoral periapical radiographs were done to evaluate bone resorption by measuring the height of the interalveolar septum of the teeth adjacent to the cleft according to the Bergland classification. Six months after surgery, graft failure occurred for 6.7% of patients who received treatment using i-PRF and A-PRF, demonstrating PCs’ ability to reduce bone resorption in comparison to the control group where graft failure occurred for 40% of patients. Additionally, periodontal status assessment of teeth adjacent to the cleft alveolus showed improvement-reducing teeth mobility and pocket depth. Furthermore, results were more prominent using autologous bone graft harvested from the iliac crest combined with PRF, which showed their ability to promote periodontal healing [36]. 

Another study done by Shawky et al., also compared the use of PRF (3000 rpm, 10 min) in combination with an autologous bone graft from the iliac crest to bone graft alone. In this study, PRF was used in two ways: PRF membranes were prepared to seal nasal mucosa and to cover bone graft, and PRF clot was cut into pieces and mixed with bone graft. Using computed tomography scans, newly formed bone quality and quantity were assessed. The study’s results demonstrated a statistically significant increase of newly formed-bone percentage in the bone graft and PRF groups (82.6% ± 3.9% and 68.38% ± 6.67%, respectively) (*p* < 0.05) [35]. Further studies in this field should be done to show the statistical reliability of such PRF applications.

### 2.4. Sinus Lift

A sinus lift is also a frequent surgical procedure of the atrophic maxilla to augment the maxillary sinus. PRF is widely used for clinical outcome improvement purposes. Some authors have described PRF’s effects when using it alone as a grafting material for maxillary sinus floor augmentation. A study by Cho et al., study had 40 patients with a lack of posterior teeth in the maxilla and a reduced residual alveolar bone height (RABH) not less than 5 mm. Full-thickness flap hydraulic transcrestal sinus lifting was done using saline injection and retrieval to separate and lift the membrane. The test group received PRF (408× *g*, 12 min) as a filler to support an elevated membrane, whereas in the control group, 5 mL saline was used. Forty-five implants no longer than 11.5 mm were placed in the posterior maxilla with RABH 6.8 ± 1.1 mm. During 12 months follow-up results were evaluated using CBCT. All implants survived. The use of saline and PRF induced an intrasinus bone gain, but the PRF filling led to a significantly greater intrasinus bone gain after 12 months (2.6 ± 1.1 mm in comparison to 1.7 ± 1.0 mm). In addition, RABH increase during the one-year follow-ups varied at different points in time, showing a higher peak immediately postoperatively and then decreasing until six months postoperatively, when it stabilized [37].

Kaarthikeyan et al., used PRF as a sole filling material in comparison to blood clot alone in maxillary sinus augmentation with the implants acting as tent poles to compare bone formation. Using a lateral bony window approach, the sinus membrane was elevated, and 10 mm implants were placed on both sides of the maxilla. On one side, the elevated sinus space was filled with PRF (2500 rpm, about 280× *g*, 10 min) [57] without mention of specific parameters, whereas on the other side, the space was allowed to fill with blood clot. A statistically significant difference in bone formation between groups were not found, but the authors concluded that PRF as a filling material was more effective because there were statistically significant differences in buccal, palatal, and average height gain between groups immediately postoperatively and 12 months later. Other parameters (mesial, mean buccal, mean palatal height differences) did not show significant differences [38].

Another study done by Olgun et al. demonstrated success in the use of titanium-prepared platelet-rich fibrin (T-PRF) (878× *g*, 12 min, titanium tubes) as the sole filling material for sinus lift surgery. The sinus lift was done using a lateral bone window approach, and a latex balloon was expanded to elevate the sinus membrane. The acquired space in the test group was filled with T-PRF and, in the control group, an allograft (CTBA Allograft, Magnesitstr, Donau, Austria). A collagen membrane (Mem-Lok: Resorbable Collagen Membrane; BioHorizons, Oakland, CA, USA) was used to close bony windows. CT was performed, and implants were installed four and six months after the sinus augmentation. The histological and histomorphometric analyses reported that the newly formed-bone ratio in the T-PRF group (16.58 ± 1.05) was lower than when using an allograft (17.28 ± 2.53), but there was no statistically significant difference between them. In addition, no significant differences were shown in cancellous bone ratios, but using T-PRF led to more cancellous bone. Overall, the use of T-PRF accelerated bone formation after four months showed comparable results to the use of an allograft for sinus lift after six months. In addition, all implants installed in both groups reached stability (measuring implant stability quotient (ISQ)) and were not statistically different (68.50 in the test and 66.37 in the control groups) [39]. The results reported in this study positioned T-PRF and allografts as comparable materials that could both be successfully used to promote bone formation.

A different PRF was also used in combination with various biomaterials for sinus floor augmentation. Pichotano et al., combined L-PRF (3000 rpm, 300× *g*, 10 min) with deproteinized bovine bone mineral (DBBM) for maxillary sinus augmentation for early implant placement. In split-mouth design research, the sinus lift was done using DBBM and DBBM with L-PRF. Afterwards, dental implants were installed in augmented sites of the maxilla after four and eight months in the test and control groups, respectively. The use of L-PRF with DBBM showed increased bone formation and allowed implant placement after four months. The study showed increased bone formation by histological evaluation when using L-PRF together with DBBM (44.58% ± 13.9%), allowing the placement of implants four months after sinus augmentation compared to the side where DBBM (30.02% ± 8.42%) was used and implants were placed eight months after augmentation. It should be noted that ISQ was significantly higher immediately after implant placement where DBBM was used (75.13 ± 5.69 in comparison to 60.9 ± 9.35); however, loading ISQ was reduced [44].

Similarly, L-PRF (400× *g*, 12 min) was also used by Nizam et al., in a split-mouth study, but it did not show improvements in bone regeneration or in various other parameters. Bone samples for histological analysis were obtained during implant placement six months after sinus lift. Several parameters evaluated in this study, such as newly formed-bone percentages, residual bone graft, bone graft in contact with newly formed bone and soft tissue, and bone height, radiographically showed similar results [40]. Gurler et al., also showed no significant improvements using L-PRF (2700 rpm, 408× *g*, 12 min) clots and membranes with allogenous freeze-dried corticocancellous bone chips (Mineross^®^, BioHorizons, Birmingham, AL, USA) in sinus lift surgery in comparison to the use of allograft alone and a resorbable collagen membrane. After surgery, several parameters were evaluated, such as pain, swelling, phonetics, sleeping, eating, daily activities, missed work, and soft tissue healing, and showed some gradual improvements that were not considered statistically significant [41]. L-PRF (3000 rpm, 10 min) as well as P-PRP were also tried in combination with a beta-tricalcium phosphate (b-TCP) graft substitute to improve bone regeneration in sinus augmentation surgery compared to b-TCP alone. Kilic et al., in this study found that adding any PCs to the b-TCP does not significantly improve bone formation and regeneration. Interestingly, despite no significant differences in histological assessment, higher capillary vessel density was found in the P-PRP with b-TCP group. In addition, L-PRF with b-TCP demonstrated a lower rate of osteoprogenitor cells and more inflammatory cells than in other groups [42].

There is also a controversial question about what kind of membrane to use in the maxillary sinus augmentation approach window. Gassling et al., evaluated the effect of a PRF membrane (400× *g*, 12 min) on bone regeneration when used to close the sinus augmentation lateral osteotomy approach site. Bilateral sinus augmentation using autologous bone and bone-substitute material (Bio-Oss) in a 1:1 ratio was covered with a PRF membrane and a conventional collagen membrane (Bio-Gide) for comparison. Implants were placed five months after surgery. Histomorphometric examination using bone samples taken during implantation showed a similar amount of vital bone (17.0% using a PRF membrane and 17.2% using a collagen membrane) and residual bone-substitute material (15.9% and 17.3%, respectively) [43]. It should be noted that what makes the use of PRF beneficial is its simple preparation and cost-effectiveness.

### 2.5. Dental Implant Placement

Dental implant placement is also an attractive field for the use of PRF. Several studies were done evaluating the PRF effect on dental implant stability. For example, Oncu et al., placed a minimum of two tapered implants (Ankylos, Dentsply/Friadent) in 20 patients using a PRF membrane (prepared at 2700 rpm, 12 min), which was applied to osteotomy sites. The obtained serum from the preparation of the PRF membranes was used to rinse implants before their placement. In control sites, no PRF was used in osteotomies or for rinsing. During the follow-up period, resonance frequency analysis (RFA) was done by presenting ISQs. While there were no differences in insertion torque values or primary stability, one week after surgery, the mean ISQ showed statistically significant differences between the PRF group 69.29 ± 10.51 and the control group 60.03 ± 12.2 (*p* = 0.002). The mean ISQs after one month were higher and showed statistically significant differences between the groups (77.19 ± 6.06 and 70.49 ± 7.74 for PRF and control groups, respectively) (*p* = 0.001). It was concluded that PRF provided faster implant osseointegration and increased implant stability during the early healing period [45]. Similarly, a beneficial effect of L-PRF (2800 rpm, 12 min) was shown by Tabrizi et al. In their study, PRF was placed in the implant site immediately before implant (BEGO Implant Systems GmbH & Co. KG, Bremen, Germany) placement in the molar region. In this split-mouth study, implants placed in the other side of the molar region without the addition of PRF were used as a control group. Two, four, and six weeks after implant insertion, ISQ showed statistically significant differences between the PRF and control groups (*p* < 0.05). It was concluded that PRF enhances implant stability during the healing period [46]. 

There are also controversial studies where the PRF beneficial effect is not unequivocally observed and did not show statistical significance. For example, Diana et al., published the results of their study where a PRF membrane prepared according to Choukroun’s protocol (3000 rpm, approximately 400× *g*, 10 min) [58] was placed in the peri-implant gap after frontal tooth extraction and immediately followed by implant (Osstem TS Implants) placement in the study group. The same surgical procedure was done in the control group but without the use of PRF. After a one-year follow-up period, the study results showed no significant difference in implant stability between groups. Furthermore, both groups had a similar increase in implant stability after a three-month time period. Fractal analysis of the bone showed a slightly lower fractal dimension (FD) in the study group, which implies more dense trabecular structures and more stable bone microstructure at sites where PRF was used. However, these FD changes between the groups were not evaluated as statistically significant. A greater increase in crestal bone height in the PRF group was found, but it had no statistically significant difference when compared to the control group [47]. In addition, Khan et al., used a PRF (3000 rpm, 12 min) membrane in immediate nanopore implant placement after tooth extraction and evaluated peri-implant tissue changes. In the study group, after atraumatic extractions and preparation of osteotomy sites, titanium implants were irrigated in PRF fluid, obtained by PRF clot squeezing, and placed following a PRF membrane placement around the implant. The control group did not receive any PRF additives. Follow-ups were done up to nine months after surgery and showed a greater mean bone loss in the control group in both the mesial and distal side of the implants when compared to the PRF group, although these differences were not considered statistically significant. In addition, peri-implant probing depth in both groups significantly increased during the follow-up period, which was more pronounced in the control group but did not show statistical significance [48]. It should be noted that the results of this study show the favorable tendency of PRF used in such a way to improve the results of implant placement. 

Another method of PRF application was described by Hamzacebi et al., who prepared PRF membranes and plugs according to Choukroun’s protocol (3000 rpm, approximately 400× *g*, 10 min) [58] and combined them with conventional flap surgery to treat periimplantitis with horizontal bone loss of 1–5 mm and infrabony components. In each patient control group, dental implants underwent only surface decontamination, whereas in the test group implants, peri-implant infrabony defects were filled with the PRF membranes, and PRF plugs were placed over the suprabony component. In this study, the PRF group demonstrated a statistically significantly higher mean probing depth (PD) reduction (*p* < 0.001) and clinical attachment level (CAL) gain (*p* < 0.01) compared to the control group three and six months after treatment. Although both treatments reduced PD, only the PRF group six months after treatment showed a statistically significantly lower PD value (3.30 ± 0.49 mm) (*p* < 0.001) than in the control group. Similarly, six months after treatment, the PRF group demonstrated a statistically significant increase in keratinized mucosa (KM) (0.62 ± 0.58 mm) (*p* < 0.001) compared to the baseline. While both treatments significantly reduced bleeding on probing (BOP), a significant difference in mean BOP scores between both treatment groups was not found. The study results indicate that using PRF in combination with flap surgery can improve the clinical efficacy of peri-implant bone defect treatment [49].

Temmerman et al. showed another interesting application of PRF using L-PRF membranes to increase the width of the KM around implants. In this split-mouth randomized control trial, the vestibular split-thickness flap in the lower jaw was made, displaced, and sutured to the periosteum, leaving the uncovered area around the implants. The test site area around the implants was covered with four mutually sutured L-PRF membranes (2700 rpm, 408× *g*, 12 min) in the study group, whereas in the control site, a free gingival graft (FGG) was used. After a six-month follow-up period, the mean gain of KM had a statistically significant increase (*p* < 0.05) in the control group where FGG was used (7.3 ± 1.2 mm) when compared to the test group (6.0 ± 0.8 mm). However, in both sites, buccolingual width of KM achieved the goal of ≥2 mm in the vestibular region of the implants. In addition, the study showed the ability of L-PRF membranes to reduce postoperative pain sensation, showing significantly lower (*p* < 0.05) pain values according to the Dutch version of the McGill Pain Questionnaire in test sites compared to control sites when measured continuously up to six days after surgery. In addition, the mean surgical time required for these surgeries was significantly lower in test sites compared to control sites (29.1 ± 4.8 min and 48.1 ± 7.7 min, respectively) (*p* < 0.05). The study showed that L-PRF membranes were effective in increasing the width of KM around implants, could shorten the surgical time, and decrease the postoperative pain level [24].

## 3. Antimicrobial Activity of PRF in Studies In Vitro

Ever since the antimicrobial efficacy of PCs was revealed, it has remained one of the most questionable topics [17]. There has been a significant increase in the number of published articles about the antimicrobial activity of PCs (using key words “platelet-rich”, “antibacterial”, and “antimicrobial”), and the considerable diversity of results in various studies makes this topic attractive (Figure 2). 

Antimicrobial activity can be defined as the destruction or inhibition of the growth of microorganisms (bacteria, fungi, and viruses) [59]. There are many pathological conditions where microbial agents have a significant or even the most important role. In the human body, wound healing is compromised by possible infection, especially in wounds in the oral region. Thus, the success of the use of PRF may be connected with its antimicrobial properties. That is why, in the last decade, numerous studies were done to find out and characterize the antimicrobial activity of PRF. Fully autologous second-generation PRF with no additives as prepared using one centrifugation may most directly reflect the antimicrobial potential of blood-derived material. 

### 3.1. Mechanisms Involved in the Antimicrobial Action of PRF

The exact component responsible for the antimicrobial activity in PCs has not yet been established but there are several theories on which mechanisms and conditions are involved according to the PCs’ content [60]. 

PCs, including L-PRF, are characterized by an increased level of platelets, which are considered cells with a multifunctional role in antimicrobial host defense [61]. According to the literature, approximately 97% of the platelets from the initial blood harvest are concentrated in PRF when prepared according to Choukroun’s protocol (3000 rpm, 10 min) [62]. First of all, platelets provide a rapid response to microbial colonization of vascular endothelium and are the earliest cells at the affected site. Platelets are able to incorporate microbes themselves and perform pathogen clearance, killing or damaging microorganisms [61]. Platelets, stimulated by microorganisms, are able to release peptides with antimicrobial properties, such as platelet factor 4 (PF-4), connective tissue activating peptide 3 (CTAP-3), thymosin beta 4 (Tβ-4,), platelet basic protein (PBP), and fibrinopeptide B (FP-B), and generate oxygen metabolites such as superoxide, hydrogen peroxide, and hydroxyl free radicals [61,63]. Platelets carry out antibody-dependent cell cytotoxicity against microbial pathogens and may modulate complement system activity by amplifying or suppressing the complement fixation [61,63].

Furthermore, in various types of PCs, there are large numbers of already well-known important immune cells: leukocytes. According to Dohan et al., more than 50% of the leukocytes from the initial blood sample are concentrated in PRF [62]. Neutrophil leukocytes contain primary and secondary granules with enclosed antimicrobial proteins, peptides, and enzymes, such as lactoferrin, defensins, bactericidal/permeability-increasing protein (BPI), azurocidin/heparin-binding protein, cathelicidins, phospholipases A2, calprotectin, etc. After activated neutrophils migrate to the site of infection, pathogens are exposed to released antimicrobial substances either in the phagosome or extracellularly [64]. Another cell type that reaches an even higher concentration than other leukocytes are lymphocytes, which are involved in innate and adaptive immune responses [62,64]. Lymphocytes are activated by bacterial products or cytokines and carry out immune responses either through mediators or directly [64].

Another system that can provide antimicrobial activity in the use of PCs is the complement system, which activates the complement cascade and promotes the lysis of microorganisms. The complement system also stimulates leukocyte recruitment for humoral defense realization [17].

### 3.2. Microorganisms Used in PRF Antimicrobial Tests

Analyzing the spectrum of bacteria used in studies, it could be claimed that PRF antimicrobial testing is mainly done on oral flora bacteria. This could be explained by the most frequent application of PRF in the craniomaxillofacial region and a wide spectrum of existing pathogens in this area. Both gram-positive and gram-negative bacteria have been used equally as often. 

Using the above-mentioned methods during the last decade, PRF antimicrobial activity is tested more often on several bacterial reference cultures and less frequently on fungal reference cultures, as shown in Table 3. 

PRF is tested less often on clinical isolates, which usually are more pathogenic bacteria and may present more clinically severe infections. Clinical isolates are usually obtained from bacterial samples taken from the oral cavity, such as biofilm-producing *Staphylococcus* species *S. aureus* and *S. epidermidis* [68]. However, some studies do not specify microorganisms but use the whole spectrum of supragingival plaque and dental root canal microflora specimens without clarification [60,69,70,71]. 

### 3.3. Antimicrobial Activity Assessment According to Methodology

As PRF is not characterized as a homogenous fluid but as an extracellular matrix with specific architecture and clotting ability, it may create difficulties in antimicrobial activity assessment using the usual methods. Antimicrobial properties of PRF have been noticed but are still open for discussion because the mechanisms responsible for such properties are still unclear [67]. Studies on PRF properties demonstrate heterogeneous results, but the last decade’s studies on PRF antimicrobial activity prove its antimicrobial properties. L-PRF membranes and exudate [17,66,67], I-PRF [60], PRF obtained by horizontal centrifugation (H-PRF) and its clot, membranes, and exudate [16], and unspecified types of PRF [65,69,71] were used to describe their antimicrobial properties. The use of the appropriate methodology for antimicrobial activity testing is crucial, and the methodology should be adjusted in order to avoid false results that could be the reason for the hitherto controversial assessment of PRF properties. Therefore, in this review, antimicrobial studies done in the last decade are briefly described and analyzed according to their used methodologies.

#### 3.3.1. Disk Diffusion Method

The disk diffusion method, or Kirby–Bauer method, is one of the most common methods used to determine bacterial sensitivity to antibiotics. In this method, sterile paper disks are impregnated with antimicrobial agents, such as PRF, and placed on bacteria-seeded agar plates. Diffusion of an antibacterial agent from the disk results in the inhibition of bacterial germination and growth, forming a zone of inhibition (ZOI) (Figure 3). The main advantages of this method are its relatively simple preparatory procedure, its low cost, and the simple evaluation of results [72,73,74]. Bacterial suspensions are usually prepared using different cultures with an optic density of 0.5, according to the McFarland standard.

The disk diffusion method is rarely described in PRF antimicrobial testing, which is probably because of the PRF’s ability to clot. Nevertheless, Feng et al. described the use of a filter paper diffusion method using exudate obtained from L-PRF (700× *g*, 12 min) and H-PRF (700× *g*, 8 min, using horizontal centrifuge). Both PRF clots were divided into five equal parts, according to layers, and drained into the exudate and solid parts. The exudate was pipetted onto pieces of filter paper and placed on agar plates for 24 h of incubation. This experiment proved the antimicrobial properties of L-PRF and H-PRF exudates against *S. aureus* (ATCC BAA-1758) and *E. coli* (MG 1655), thus marking the properties of PRF liquid components. Exudates from both PRF clots were more active against *E. coli.* H-PRF exudate showed statistically significantly higher bacterial inhibition rates demonstrating better antibacterial activity compared to L-PRF exudate.

#### 3.3.2. Agar Diffusion Method

The agar diffusion method is similar to the disk diffusion method, but a tested antimicrobial agent, such as PRF, is placed directly on the agar plate surface [75]. Bacterial suspensions are usually prepared using different cultures with an optic density of 0.5, according to the McFarland standard. This methodology was used in a study by Castro et al., where an L-PRF membrane (408× *g*, 12 min) and L-PRF exudate (obtained after L-PRF clot formation) were used to characterize antimicrobial properties. The L-PRF membrane demonstrated a visible antibacterial effect against periodontal pathogens, as shown in Table 4. 

*P. gingivalis* reached statistically significantly larger inhibition in comparison to other bacteria (*p* < 0.05). Furthermore, subtracting changes in membrane size that appeared during 72 h of incubation from mean ZOI, only *P. gingivalis* showed a statistically significant mean ZOI of 9.1 ± 3.2 mm^2^ (*p* < 0.05). L-PRF exudate demonstrated inhibition (mean ZOI 17 ± 2.6 mm^2^) only against *P. gingivalis* [66]. Similarly, Melo-Ferraz et al. used an L-PRF membrane (2700 rpm, 408× *g*, 12 min) to assess antibacterial activity against *E. faecalis* (ATCC 29212), *P. aeruginosa* (ATCC 27853), and *C. albicans* (ATCC 90028). L-PRF circular membranes measuring 6 mm were placed on agar plates and incubated for 24 h at 35–37 °C. The L-PRF membrane showed inhibitory action and proved its antibacterial effect based on a resulting ZOI between 9 and 19 mm. The results were even more pronounced than the positive control of chlorhexidine 0.12% (ZOI between 0 and 12 mm) [67]. 

In addition, a study comparing the antibacterial activity of L-PRF and H-PRF was done. Feng et al., described the inhibition of *S. aureus* (ATCC BAA-1758) and *E. coli* (MG 1655) by compressing L-PRF (700× *g*, 12 min) and H-PRF (700× *g*, 8 min, using horizontal centrifuge) clots and incubating whole PRF clots on bacterial culture plates for 24 h. H-PRF demonstrated a significantly larger-width ZOI for both bacteria compared to L-PRF. Inhibition against *E. coli* was statistically significantly more pronounced (*p* < 0.05). The authors also did L-PRF and H-PRF antimicrobial activity characterization in different layers. PRF clots were divided into five equal parts and drained in exudate and a solid part. For the exudate’s antibacterial testing, the authors used the disk-and-filter paper diffusion method described previously. The membrane, made from a solid part by compression, was placed on an agar plate for 24 h incubation. The authors found that the PRF membranes as well as exudates presented antimicrobial activity. L-PRF and H-PRF membranes showed higher activity against *E. coli* than *S. aureus*. In addition to the effect of centrifugation parameters on antimicrobial activity in this study, by increasing the count of immune cells, the comparison of whole and solid PRF clots was done, highlighting the importance of liquid PRF components [17]. 

In contrast to the previous study, no antibacterial activity from the PRF membrane (3000 rpm, 10 min), which was placed on the surface of an agar plate, was found against *P. gingivalis* (ATCC 33277) or *A. actinomycetemcomitans* (ATCC 43718) in the study done by Badade et al. In this experiment, the PRF membrane’s antibacterial activity was compared to PRP. While the PRF membrane showed no ZOI when using the agar well diffusion method, PRP statistically significantly inhibited *P. gingivalis* and *A. actinomycetemcomitans* (*p* < 0.05) [65].

#### 3.3.3. Agar Well Diffusion Method

The agar well diffusion method is one of the antimicrobial diffusion methods, but it has some technical differences from the others. An antimicrobial agent, such as PRF, is placed in wells that have been cut out in the agar [75]. Bacterial suspensions are prepared using different cultures with an optic density of 0.5, according to the McFarland standard. This methodology was used in several studies. 

Nagaraja et al. used PRF (400× *g*, 15 min) against dental root canal microflora and *C. albicans*. Forty-eight hours after PRF placement into incubation wells, the incubation results were evaluated. PRF was able to inhibit the growth of bacteria and demonstrated a mean ZOI 4.25 ± 0.88, whereas inhibition of *C. albicans* was low (mean ZOI 1.50 ± 0.53). The antibacterial activity of PRF was compared to that of platelet-rich fibrin matrix (PRFM), which is a PC with a dense and elastic fibrin matrix. PRFM is obtained using two sets of centrifugations with higher gravitational force and the use of calcium chloride. Nevertheless, a statistically significant larger mean ZOI was demonstrated using PRF (*p* = 0.00) [71]. The antibacterial activity levels of different types of PRF were also evaluated and compared to PRP in a study done by Kour et al., I-PRF (700 rpm, 3 min), PRF (3000 rpm, 10 min), and PRP were tested against *P. gingivalis* (ATCC 33277) and *A. actinomycetemcomitans* (ATCC 43718). All PCs demonstrated certain growth inhibition of *P. gingivalis* and *A. actinomycetemcomitans* with pronounced ZOIs. The mean ZOIs of i-PRF were 15.2 ± 5.7 and 9.8 ± 1.75, the mean ZOIs of PRF were 8.8 ± 1.0 and 10 ± 1.3, and the mean ZOIs of PRP were 12.9 ± 4.07 and 12.5 ± 2.07, respectively. The I-PRF mean ZOI against *P. gingivalis* was wider than the mean ZOIs of PRF and PRP; however, it reached statistical significance only compared to PRF (*p* < 0.05). Nevertheless, PRP was statistically significantly more active against *A. actinomycetemcomitans*, showing a wider mean ZOI than I-PRF and PRP (*p* < 0.05) [60].

Mamajiwala et al., tested differently prepared PRF from different donor age groups against supragingival plaque flora and claimed that PRF properties, including antimicrobial activity, are highly influenced by the PRF preparation technique, including factors such as centrifugation speed, centrifugation time, relative centrifugal force expressed as revolutions per minute, and g-force, and patient factors such as patient age. In this study, PRF membranes were prepared using different centrifugation speeds (1400 rpm, 2800 rpm, 3500 rpm) and times (8 and 15 min) within the relative centrifugation force spectrum of 228–1425× *g*. PRF samples were placed in prepared agar wells. Antimicrobial activity was evaluated by analyzing the mean ZOI of the tested biomaterial. The results showed that the highest antimicrobial activity was found in PRF membranes prepared at the following parameters: 1400 rpm, 228× *g*, and 8 min. PRF prepared in this way demonstrated a statistically significant wider mean ZOI in all age groups (20–34, 35–49, 50–65 years) (*p* < 0.01). Interestingly, the widest ZOI was seen in the younger age group, whereas the narrowest ZOI was seen in the 50–65 age group (*p* < 0.01) [69]. 

#### 3.3.4. Microdilution Method

The microdilution method is a classic way to determine bacterial susceptibility against antibiotic agents; however, it is used less often due to more-complicated preparation. The main advantage of this method is the ability to determine minimal inhibitory concentration (MIC) and minimal bactericidal concentration (MBC) values (Figure 4). 

Balouiri et al. defined MIC as the “lowest concentration of the assayed antimicrobial agent that inhibits the visible growth of the tested microorganism” [74]. MBC is the lowest concentration of the assayed antimicrobial agent needed to kill 99.9% of microorganisms, which is possible to determine after an additional cultivation method [75].

Castro et al., used a microdilution test to detect antibacterial activity against *P. gingivalis* and *A. Actinomycetemcomitans.* It was found that L-PRF exudate can decrease the growth of *P. gingivalis* (ATCC 33277) depending on the L-PRF exudate dose, showing a reduction of the mean growth of bacteria at 86% (*p* < 0.001), 38% (*p* < 0.05), and 24% (*p* > 0.05) when exposed to L-PRF in ratios 1:1, 1:2, and 1:4, respectively. On the contrary, *A. Actinomycetemcomitans* (ATCC 43718) demonstrated a mean increase of bacterial growth at the same concentrations of L-PRF exudate (*p* < 0.05), which can be explained by the inhibition of bacterial autoaggregation [66]. Jasmine et al. presented an evaluation of the antimicrobial activity and antibiofilm effects of i-PRF (1000 rpm, 5 min) against oral pathogenic biofilm-producing *Staphylococcus* bacteria. In the study, *S. aureus* and *S. epidermidis* isolated from patients with dental and oral abscesses, *S. epidermidis* reference cultures which were biofilm-forming, (ATCC 35984) and biofilm-negative strains (ATCC 12228) as negative controls were tested. Isolated bacterial strains were determined as moderate producers of biofilm for *S. epidermidis* and strong producers of biofilm for *S. aureus.* Using a microdilution method, it was found that, at an MIC value of 80 μL/mL, i-PRF could inhibit the growth of nonbiofilm-producing bacteria, and at an MBC value of 160 μL/mL, it showed bactericidal activity. For biofilm-producing bacteria, the MIC and MBC values of i-PRF were higher at 160 μL/mL and 240 μL/mL, respectively [68].

#### 3.3.5. Plate Counting Assay

In a study done by Feng et al., the plate counting assay was described as an antibacterial testing method for liquid PRF. The liquid states of L-PRF (700*× g*, 12 min) and H-PRF (700*× g*, 8 min, using horizontal centrifuge) were separated into five different layers by sequential pipetting. Each obtained liquid layer was mixed in equal concentrations with 100 μL bacterial suspensions of either *S. aureus* (ATCC BAA-1758) or *E. coli* (MG 1655) (1 × 10^6^ CFU·mL^−1^). After four hours of incubation at 37 °C, samples were diluted in PBS. Colony-forming units (CFU) were counted after final overnight culturing on agar plates. The study showed significantly different patterns between both PRF liquids’ antimicrobial activity within their layers. While L-PRF’s deepest layer showed more pronounced antimicrobial activity, H-PRF’s layers did not demonstrate any significant differences among each other [17].

#### 3.3.6. Biofilm Inhibition Assay

Only a few methods are described to test the activity of PRF against biofilm. A suitable method to evaluate PRF’s effects on biofilm is the semiquantitative plate method. After incubation of bacteria in 96 well plates, an antimicrobial agent is added to the wells and further incubated. After washing off the nonadherent cells and fixation of adherent cells in the biofilm, the wells are strained, washed, and dried. The quantity of biofilm is evaluated using light absorbance. By using the described method, Jasemine et al. proved i-PRF’s (1000 rpm, 5 min) antibiofilm potential by its reduction of biofilm formation at an MIC of 160 μL/mL against weak, moderate, and strong biofilm-producing *S.epidermidis* strains. The reductions were 30.4 ± 2.17% for the weak strain, 36.2 ± 2.52% for the moderate strain, and 49.1 ± 8.79% for the strong strain (*p* < 0.005). At an MBC of 240 μL/mL, all biofilm-producing strains showed an inability to form a biofilm [68].

A specific method of evaluation of PRF activity against biofilm is described by Sanchez et al., In this study, L-PRF exudate’s effects against *P. gingivalis* were tested either in multispecies preformed biofilm or in biofilm during its formation. A 14-species community was preformed using a BIOSTAT B TWIN bioreactor (Sartorius, Germany). L-PRF did not show a statistically significant decrease of *P. gingivalis* concentration in preformed multispecies biofilm, but in developing multispecies biofilm, it was able to better decrease *P. gingivalis* concentration in comparison to PBS (*p* = 0.006), inactive L-PRF exudate, and L-PRF exposed to horseradish peroxidase [76].

## 4. Immunological Activity of PRF

Another very important and, at the same time, basic property of PRF is its ability to release various growth factors and cytokines to stimulate tissue regeneration [66]. For medical applications, PRF has garnered attention as a beneficial tool due to its ability to deliver autologous growth factors and cytokines in supraphysiologic concentrations topically. Various studies have been done in the last decade to clarify this very promising feature, reach a better understanding, and identify influencing factors to be able to explain different results regarding the use of PRF.

### 4.1. PRF Influence on Living Cells In Vitro

Several authors have investigated PRF’s influence on living cells and described its stimulating effect. The ability of PRF to promote bone regeneration is described in a study done by Kim et al., who applied PRF (10*× g*, 10 min) on osteoblast cell cultures from iliac crest bone marrow and showed its ability to promote proliferation and differentiation. It was found that 3% PRF (prepared in Dulbecco’s Modified Eagle Medium (DMEM)) increased osteoblast cell DNA synthesis in 24, 48, and 72 h, thus increasing osteoblast proliferation and showing statistically significant differences compared to 10% normal human serum (NHS) and 10% fetal bovine serum (FBS), both of which were prepared in DMEM (*p* < 0.05). Analyzing osteoblast differentiation and anticancer activity, a 3% PRF application demonstrated an increase in protein synthesis in 24, 48, and 72 h, and it showed differences compared to 10% NHS and 10% FBS. Another finding was the increased alkaline phosphatase (ALP) activity of osteoblasts in 24, 48, and 72 h due to 3% PRF application as compared to 10% NHS and 10% FBS. During the calcification process, ALP transports inorganic phosphate, ensures cell division, and regulates differentiation. It can also act as a marker for bone cell differentiation. PRF showed a higher amount of such growth factors as platelet-derived growth factor (PDGF) and transforming growth factor beta (TGF-β) than in NHS (*p* < 0.05), which stimulates the before-mentioned osteoblastic response and thus predicts stimulated bone regeneration and wound healing in vivo as well [77]. Another crucial factor of PRF is its angiogenic activity, which is already well-known but described in detail by Vizcaino et al. Their study shows the effect of PRF (prepared at 600 rpm, 44*× g*, 8 min, and 2400 rpm, 710*× g*, 8 min) on human dermal vascular endothelial cells (HDVECs) and human primary fibroblasts (HPF) [78]. In addition, a remarkable effect of PRF was described in a study done by Mudalal et al. by demonstrating the effect of L-PRF (3000 rpm, 1278*× g*, 12 min) on proinflammatory cytokine release. The authors described a different concentration of interleukin 1 beta (IL-1β), tumor necrosis factor-alpha (TNF-α), and interleukin 6 (IL-6) in the culture medium (DMEM supplemented with 10% of fetal bovine serum) of human gingival fibroblast cells (HGFCs) treated with 1 μg/mL lipopolysaccharide from *P. gingivalis* (PG-LPS) and HGFCs treated with PG-LPS (1 μg/mL + L-PRF) after 20 h of incubation. L-PRF showed the ability to suppress the secretion of proinflammatory cytokines (*p* < 0.001) [79].

### 4.2. Growth Factor and Cytokine Release Pattern of PRF In Vitro

There are numerous studies describing the release of various growth factors and cytokines from PRF as well as comparisons with other PCs. As mentioned before, PRF can cause remarkable changes in cells in vitro, but clinical efficacy is still considered controversial. The reason for such variable results is the differing content of PRF between studies, which depends not only on patient-specific factors, such as general health condition and age, but is also strongly affected during the preparation of the biomaterial through various centrifugation protocols and centrifugation systems. Several authors described the effect of different centrifugation protocols on growth factor and cytokine release and confirmed the protocols’ strong impact. It is also proven that it is not only within different PCs, but also in each type of PRF, that there are different patterns of growth factors and cytokine release [80,81,82]. 

Dohan et al., described growth factor and cytokine release from L-PRF (2700 rpm, around 400*× g*, 12 min) and A-PRF (1500 rpm, 14 min) membranes. L-PRF rapidly and increasingly released TGFβ-1, PDGF-AB, and vascular endothelial growth factor (VEGF) during the first 24 h and, for seven days further, continued to release significant amounts of molecules at a slower pace, although A-PRF was able to release molecules overall only for three days until complete dissolution of the membrane. Nevertheless, both PRF types demonstrated slow-release kinetics compared to PRP. L-PRF showed a statistically significant increase in growth factor release level (*p* < 0.001) at all points in time in comparison to A-PRF [81]. 

Similarly, Koboyashi et al., analyzed the release of growth factors from PRF (2700 rpm, 325*× g*, 12 min) and A-PRF (1500 rpm, 100*× g*, 14 min) clots and also compared them to PRP (100 rpm, 45 g, 7 min, further 3000 rpm, 400*× g*, 10 min). Significant differences in growth factor release kinetics were found: PRF demonstrated a continuous and stable release of growth factors during a 10-day time period, releasing more growth factors at later points in time in contrast to PRP’s early, high, and rapid release. Such a localized and sustained release of growth factors enriches the environment with signaling molecules for a longer period, thus providing stronger induction of the cell proliferation, migration, and differentiation necessary for achieving optimum tissue regeneration. In addition, the total amounts of growth factor released from different PCs during the 10 days were different. A-PRF demonstrated much higher average levels overall regarding PDGF-AA, PDGF-AB, TGF-β1, and epidermal growth factor (EGF) [82]. 

Another author, Jasmine et al., described the cytokine expression pattern of L-PRF (2700 rpm, 12 min) and i-PRF (1000 rpm, 5 min) and found remarkable differences in the cytokine quantity released from T helper 1, T helper 2, and T helper 17 between both types of PRF. First, the cytokine pattern in both PCs was found to be different. Whereas significantly higher levels of IL-1α, interleukin-2 (IL-2), IL-6, and interleukin-8 (IL-8) were found in L-PRF, i-PRF demonstrated abundant expression of interleukin-4 (IL-4), IL-6, and interleukin-10 (IL-10). However, overall, L-PRF demonstrated higher cytokine levels than i-PRF [80]. 

In a study done by Dohan et al. one of the analyzed molecules released from PRF was bone morphogenetic protein 2 (BMP 2). Interestingly, BMP 2 was released only from an L-PRF (2700 rpm, around 400*× g*, 12 min) membrane, but release from an A-PRF (1500 rpm, 14 min) membrane was not detected. Molecules were continuously released from L-PRF cells but not, however, released from platelets. This points out the importance of the population of other cells within PRF [76]. Vizcaino et al., described the PRF effect in vitro on the monoculture and coculture of human dermal vascular endothelial cells (HDVECs) with human primary fibroblasts (HPF) and showed not only different growth factor release amounts and patterns, but also demonstrated different proangiogenetic effects depending on PRF type (centrifugated at 2400 rpm, 8 min, 710*× g* or 600 rpm, 8 min, 44*× g*). PRF prepared using low centrifugation force not only demonstrated a significantly greater release of growth factors, but also caused the formation of significantly more microvessel structures [78].

Taking into consideration the mentioned differences in PRF content, the question regarding its function is apparent. An exciting discovery of Jasmine et al., is a confirmation of different PRF actions through diverse pathways. The authors identified molecular interactions of cytokines and growth factors using the Retrieval of Interacting Genes/Protein database (STRING). It was shown that L-PRF (2700 rpm, 12 min) and i-PRF (1000 rpm, 5 min) work through different signaling pathways (NF-k B, Toll-like receptor signaling, MAPK signaling via T-cell receptor signaling pathways, and JAK-STAT signaling pathways, respectively), thus highlighting their functional diversity [80]. 

### 4.3. PRF Cell Quality

Another important factor affecting the quality of PRF and its role in tissue regeneration is the state of the cells’ condition in the biomaterial. Activated cells in PRF have higher potential for tissue regeneration [81]. As PRF is a blood-derived biomaterial containing living cells, it should be carefully prepared and kept under proper conditions to keep cellular content alive [83]. Cell quality in L-PRF was demonstrated in a study done by Dohan et al., where L-PRF was prepared using four different centrifuges. Centrifugation force was set at 400*× g*, and rotational speed (rpm) was adjusted corresponding to the Intra-Spin centrifuge, as shown in Table 5.

Scanning electron microscopic evaluation of L-PRF membranes prepared using the Intra-Spin centrifuge showed the observed cells had a normal shape and were alive, besides lymphocytes that presented a textured surface, indicating activated cells [81]. In contrast, all cells in L-PRF membranes prepared using other centrifuges were shrunken or squashed, and no cells were found in a normal or activated shape. The reason for such cell damage is described as the high intensity of vibrations during the centrifugation process. A rise in radial vibration above one can cause resonance in centrifugation tubes, and it may result in additional mechanical influences on cells and cause cell damage. Another possible reason for cell damage could be a rise in temperature during the centrifugation process. In the same study, a comparison between the temperatures of the surface at the center of the tubes before and after centrifugation using an infrared thermometer was done, and it revealed a rise in temperature during the centrifugation, which varied between the four tested centrifuges. The centrifugation machine (Intra-Spin) that provided the lowest temperature also showed the best cell quality in PRF [81]. Based on the mentioned findings, it can be concluded that PRF, as a complex autologous system containing cells and proteins, is highly dependent on the specific preparation protocol and methodology, thus determining the biological potential, function, and clinical significance of PRF. 

### 4.4. Immunological Effect of PRF in Humans

Proving immunological action by showing continuous growth factor release of PRF in vivo is complicated due to the possibilities in evaluating growth factor release. Only one study was found to describe growth factor continuous release in gingival cervical fluid. Gamal et al. used a xenograft material on its own and in combination with PRF (2500 rpm, about 280*× g*, 10 min) [57] and PRGF to treat periodontal intrabony defects. The materials were also covered with PRF and PRGF membranes, respectively. To evaluate released growth factors (VEGF, PDGF-BB), the gingival cervical fluid was collected at different periods up to 30 days after therapy. The study showed that the highest mean PDGF-BB concentration in cervical fluid in all study groups except growth factor concentration reduction was found to be slower in cases where PCs were used, although the differences between these three groups were insignificant. In addition, VEGF levels showed a gradual decrease with no statistical difference between the groups. Despite the beneficial PRF characteristics when used in vitro, the treatment mainly was based on the use of a xenograft because its application did not lead to clinically significant differences [84]. After this finding, the PRF growth release pattern in vivo still remains questionable due to the lack of studies in other sites of the human body. It should be noted that, as mentioned by the authors, a periodontal intrabony defect is a special condition due to its open contaminated nature, continuous bacterial colonization, and presence of gingival cervical fluid. Thus, such a media could not provide maintenance of molecules in a supraphysiologic amount. This fact could explain the lack of beneficial treatment results in this clinical study using PRF [84].

## 5. Discussion

In recent years, significant steps forward have been taken in the field of biomaterial development to solve the problem of impaired tissue healing and regeneration. The last 10 years have been filled with numerous studies in the field of regenerative medicine and tissue engineering. The development of PRF has intensified, but, still, there is no clarity regarding the properties of this autologous biomaterial.

Despite widening the scope of PRF application to plastic and reconstructive surgery, traumatology, orthopedics, and aesthetic medicine, PRF is still most commonly used in oral and maxillofacial surgery. This area most often shows beneficial clinical results using PRF because of already existing experience regarding its uses or more suitable regions of application. Numerous clinical studies prove that PRF affects soft tissue and has the ability to accelerate the closure of extraction sockets as well as improve periodontal status (PD reduction, CAL gain, KM width increase) [47,59]. PRF demonstrates a reduction of postoperative pain sensation [30,31,47], swelling [23,30], and faster neurosensory recovery [55] after surgery. Acting on hard tissue, PRF is able to reduce bone resorption [23,27,35,52], increase and accelerate bone formation [29,37,52,56,57], enhance the stability of bones after surgery [54], and provide faster dental implant osseointegration as well as increased dental implant stability, especially in early phases [42,43]. It should be noted that, despite not having superior results when comparing PRF with other biomaterials and other treatment methods, this material proved to be safe, comparatively simple to prepare and apply, and cost-effective. The reasons why PRF’s effects could be considered controversial are various, starting with incorrect material preparation, choosing the wrong centrifugation parameters, low-quality centrifuges, and finishing with an application at inappropriate sites where PRF undergoes high loading forces and cannot remain in place. PRF should not be considered a miracle solution, nor should it be expected to function under inappropriate conditions, but rather as localized natural support that induces a physiological tissue response that leads to improved wound healing and tissue regeneration. PRF mimics the physiology and immunology of wound healing [5]. 

A biomaterial that contains living cells is very unique and, at the same time, a fragile material to work with. Dohan et al. note the necessity to handle PRF carefully to keep the cells within it alive [62]. It could be influenced at any time during preparation and also during clinical application, and thus it is necessary to respect and carefully follow all recommendations during work with PRF, choose the best available technical equipment, and follow and rely on good-quality studies done in this field. PC classification is often not correctly used, leading to attribution of PRF’s definition to other PCs, such as platelet-rich fibrin matrix (PRFM), PRGF, and Vivostat PRF, which all present different properties and clinical actions, thus causing misunderstandings. In addition, often research does not provide the used methodology in details by not mentioning all centrifugation parameters and used centrifuges. An important inaccuracy factor is also the frequent lack of precise information regarding the types of collecting tubes used, whether they are made from glass, titanium and plastic, polypropylene, polystyrene, or other polymers, which can influence the coagulation process. When using glass tubes, activation of the platelets and coagulation is found to be faster because of natural coagulation-inducing silicon oxide ions, which are not present in plastic tubes [85].

This review describes several studies aimed at identifying the antimicrobial properties of PRF that have been observed but still are open for discussion. Antimicrobial activity can provide protection against microbes or treatment of various pathologies in which microbial agents are involved, thus facilitating other PRF properties to manifest. There is great importance in using the appropriate methodology for antimicrobial activity testing because of PRF’s structural complexity. Most of the used antimicrobial testing methods are based on antibacterial agent diffusion ability in culture media. However, PRF’s inhomogeneous nature, regarding the specific architecture of its extracellular matrix and its clotting ability, may interfere, thus resulting in less-reliable or heterogeneous results. Thus, to obtain more-reliable results, it is necessary to choose more-suitable methods and to make minor methodological adaptations of existing standardized methods to resemble the antimicrobial effect in vivo of any natural product, including PRF [74]. The human oral cavity contains more than 700 species of microorganisms, which makes it one of the most complex microbiomes [86]. According to analyzed studies used in this review, evaluation of the effect of PRF prevails in the oral microbiome because it is the most frequent area of PRF application and an area at high risk of infection after any surgical intervention. PRF’s antimicrobial activity was tested against periodontal pathogens *P. gingivalis* [60,65,66], *A. actinomycetemcomitans* [60,65], *F. nucleatum* [66], and *P. intermedia* [15] as well as against potential respiratory pathogens *S. aureus, E. coli* [17], and *P. aeruginosa* [60,65], which can be found in the oral cavities of patients suffering from periodontal disease. By analyzing the spectrum of bacteria used in the reviewed studies, we can conclude that both gram-positive and gram-negative bacteria have been used equally as often. Therefore, the results cannot be representative of other areas of the body. There is a lack of studies which have used clinical isolates to identify the influence of PRF, although clinical isolates may be more pathogenic than standard reference cultures.

PRF’s effect on cells was shown to be not only stimulating by promoting proliferation and differentiation of osteoblast cells [77], but also able to suppress the secretion of proinflammatory cytokines [79]. PRF demonstrates a slow, continuous, and stable release of growth factors compared to PRP [81,82]. Different PRF types showed differences not only in cytokine expression pattern [80], angiogenetic activity [78], and ability to release BMP2, but also by working through different signaling pathways [80], thereby emphasizing the importance of different types of PRF.

According to analyzed studies where PRF demonstrated antibacterial activity against different bacterial species, due to its ability to maintain a sustained release of growth factor and cell activity on material application, accurately and correctly prepared PRF can ensure antibacterial and anti-inflammatory properties. It is recommended to apply PRF in oral and maxillofacial surgery to enrich the environment with antibacterial, anti-inflammatory, and stimulating agents to achieve optimal tissue regeneration, faster patient recovery, and higher patient satisfaction. Different cytokine and growth factor release proportions from different PRF types make it an even more exciting biomaterial as it shows its influenceable nature, which can be used in further studies to improve its properties and widen its successful applications in medicine.

PRF is a material with great potential to be used in further studies to find out and properly explain mechanisms that are involved in its antimicrobial and anti-inflammatory effects. Challenging research directions include standardization for clinical outcome prediction of such a patient-specific material. Furthermore, the material has the potential to be used and studied as a drug carrier or drug delivery system and to be incorporated into porous materials for bone tissue regeneration. 

## Figures and Tables

**Figure 1 ijms-24-01073-f001:**
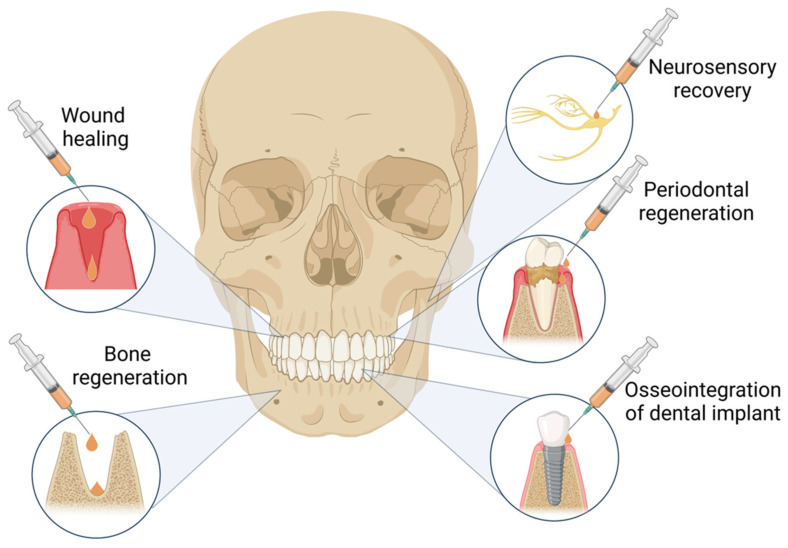
PRF application in oral and maxillofacial surgery. Figure created with Biorender.com.

**Figure 2 ijms-24-01073-f002:**
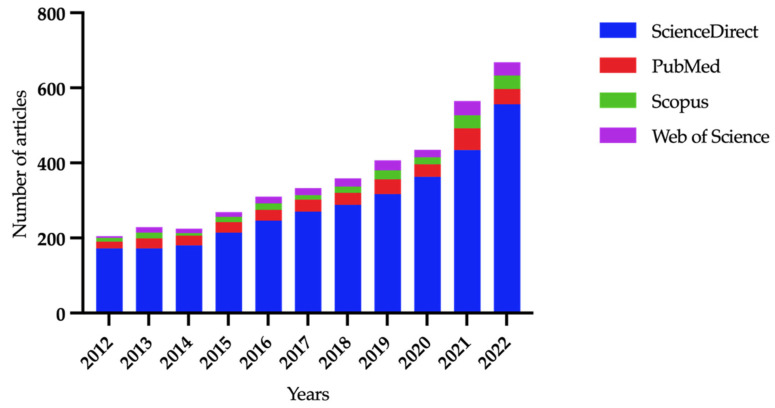
PC-related articles during the last decade in four databases: ScienceDirect, PubMed, Scopus, and Web of Science.

**Figure 3 ijms-24-01073-f003:**
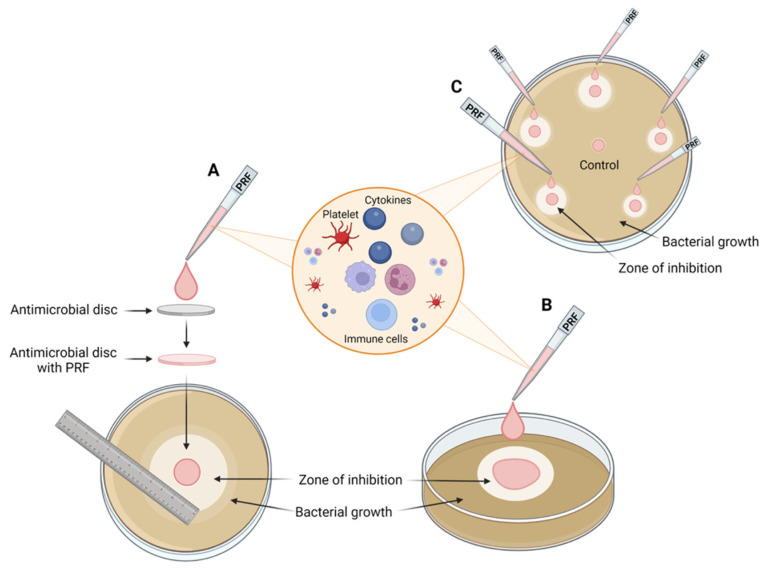
Diffusion methods for in vitro evaluation of antimicrobial activity: (**A**) Disk diffusion method; (**B**) agar diffusion method; (**C**) agar-well diffusion method. All figures created with Biorender.com.

**Figure 4 ijms-24-01073-f004:**
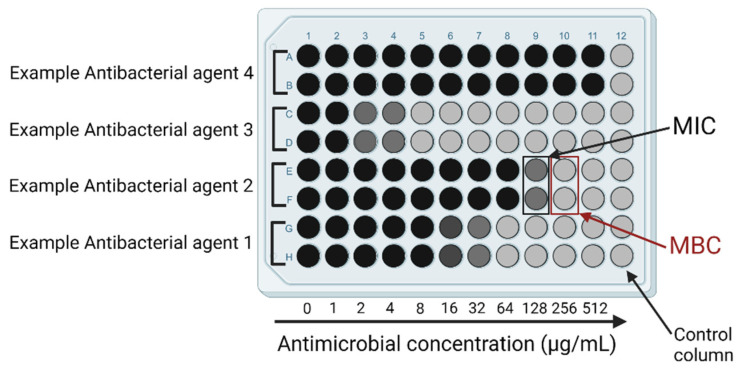
Microdilution method for in vitro evaluation of antimicrobial activity. Figure created with Biorender.com.

**Table 1 ijms-24-01073-t001:** Types of platelet-rich fibrin [12,13].

Type of PC	Explanation
Pure Platelet-Rich Plasma (P-PRP)	Preparation without leukocytes and with a low-density fibrin network after activation
Leukocyte and Platelet-Rich Plasma (L-PRP)	Preparation with leukocytes and with a low-density fibrin network after activation
Pure Platelet-Rich Fibrin (P-PRF)	Preparation without leukocytes and with a high-density fibrin network
Leukocyte and Platelet-Rich Fibrin (L-PRF)	Preparation with leukocytes and with a high-density fibrin network

**Table 2 ijms-24-01073-t002:** Classification of L-PRF derivates [13,16].

L-PRF Derivates	Centrifugation Parameters (Revolutions per minute/min)	Type of Tubes	Condition
Advanced Platelet-Rich Fibrin (A-PRF)	1300 rpm/14 min	Glass-based vacuum tubes	Gel form
Advanced Platelet-Rich Fibrin + (A-PRF +)	1300 rpm/8 min	Glass-based vacuum tubes	Gel form
Injectable Platelet-Rich Fibrin (i-PRF)	700 rpm/3 min	Plastic-based vacuum tubes	Liquid form
Injectable Platelet-Rich Fibrin Male (i-PRF M)	700 rpm/4 min	Plastic-based vacuum tubes	Liquid form
Injectable Platelet-Rich Fibrin + (i-PRF +)	700 rpm/5 min	Plastic-based vacuum tubes	Liquid form
A-PRF Liquid (A-PRF (L))	1300 rpm/5 min	Glass-based vacuum tubes	Liquid form

**Table 3 ijms-24-01073-t003:** Reference cultures of bacteria and fungi used in studies.

Bacteria and Fungi	Reference Culture Number	References
*Staphylococcus aureus*	ATCC BAA 1758	[17]
*Escherichia coli*	MG 1655	[17]
*Porphyromonas gingivalis*	ATCC 33277	[60,65,66]
*Prevotella intermedia*	ATCC 25611	[66]
*Pseudomona aeruginosa*	ATCC 27853	[67]
*Aggregatibacter* *actinomycetemcomitans*	ATCC 43718	[60,65,66]
*Fusobacterium nucleatum*	ATCC 20482	[66]
*Enterococcus faecalis*	ATCC 29212	[67]
*Candida albicans*	ATCC 90028	[67]

**Table 4 ijms-24-01073-t004:** Antibacterial effect of periodontal pathogens [66].

Bacteria	Reference Culture	Mean ZOI
*P. gingivalis*	ATCC 33277	11.8 ± 5.0 mm^2^
*P. intermedia*	ATCC 25611	2.7 ± 5.2 mm^2^
*F. nucleatum*	ATCC 20482	2.6 ± 3.0 mm^2^
*A.Actinomycetemcomitans*	ATCC 43718	0.6 ± 1.7 mm^2^

**Table 5 ijms-24-01073-t005:** Centrifuges and centrifugation parameters used to prepare L-PRF [79].

Centrifuge	Centrifugation Force	Rotations per Minute	Time
Intra-Spin	400*× g*	2700 rpm	12 min
A-PRF machine	400*× g*	2400 rpm	12 min
LW centrifuge	400*× g*	2300 rpm	12 min
Salvin centrifuge	>400*× g*	3400 rpm	12 min

## Data Availability

Not applicable.

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
