# Peer review of "Can Our Blood Help Ensure Antimicrobial and Anti-Inflammatory Properties in Oral and Maxillofacial Surgery?"

_ijms, 2023, doi:10.3390/ijms24021073_

Round 1

Reviewer 1 Report

Ref. No.: ijms-2036785

Subject: Decision on Manuscript: Can our blood help to ensure anti-microbial and anti-inflammatory properties in oral and maxillo-facial surgery?

Journal: Materials

Dear Editor,

I would like to thank you for the invite to collaborate to review process of article “Can our blood help to ensure anti-microbial and anti-inflammatory properties in oral and maxillo-facial surgery?”. I recommend that is necessary a major revision of manuscript. Some comments are described below:

English should be improved in all manuscript.

Abstract: The abstract should be rewritten considering the steps below, as this a review manuscript:

“This part should start by e.g. " The major points are 1)…, 2)…". Indeed you should give here the major points and advances that you demonstrate in the sections by literature analysis. Those points should be precise trends (increase, decrease…) supported by data (%, numbers), whenever possible. A convenient way to build this second part of the abstract is to write at the end of each article section a conclusion of about 1-2 sentences to summarize the major point of the section and its significance. Then all those 1-2 sentence

conclusions can be gathered in the second part of the abstract. In other words this second part of the abstract should clearly show the added value of your analysis. This second part of the abstract is indeed the solution to the problems explained in the first part. Overall it is the contrast between the first and second part that makes the value of the article for the reader, and in turn the impact in terms of citations.”

All tables and figures should be mentioned the copyrights or similar, when there are not the own authors, even there are adapted.

Line 130: “PRF use is described in alveolar ridge preservation,131 orthognathic surgery, cleft lip and palate surgery,…” References???

Line 138: “Majority of studies confirm clinical advantages of PRF, thus adeeper138 understanding of its use towards several applications is a must.” This paragraph should be joined to the before.

Line 246: “Despite that PRF is rarely described as a biomaterial foruse247 in orthognathic surgery, in some researches it demonstrated a favorable effect” References????

Line 319: “Sinus lift is also frequent surgical procedure of the atrophic maxilla toaugmentthe310 maxillary sinus. PRF is widely used for a clinical outcome improvement purpose. 311 Some authors described the PRF effect using it alone as a….” These two paragraphs should be joined.

Line 487: “There is a significant increase in the number of published488 articles about antimicrobial activity of PCs and considerable diversity of results in various studies makes this topic attractive” The authors should be included a graphical with the latest research in this area using the Web of Science, Scopus, Science Direct with source.

Line 539: “Using above mentioned methods more often PRF antibacterial activity are tested on bacterial reference cultures, specifically – Staphylococcus aureus (ATCCBAA1758),541 (ATCC 25923), Escherichia coli (MG 1655) [16], (ATCC 25922)….” A table should be included with these information’s.

Section 3.3: References????

Line 580: “The disc diffusion method is rarely described in PRF antimicrobial testing, probably581 because of the PRF ability to make clot.” Reference?

Similar to the update before, at section 3.3.2, the information about ATCC should be included in a table.

Line 717: “Only few methods are described to test the activity of PRF against biofilm formation” If there are few, then, all them should be mentioned in the manuscript.

Line 729: “Another very important and at the same time basic property of PRFis itsabilityto730 release various growth….”Reference??

In section 4.2, the first paragraph is too long.

Line 837: “Scanning electron microscopic evaluation of L-PRF membranes prepare during the Intra-Spin centrifuge….”References???

A mixed of discussion a long of text should be included, not separated.

In general, the authors emphasized the PRF in all dental applications. Nevertheless, the proposal of this review is emphasized the use of PRF as antimicrobial and anti-inflammatory agent, and these are not the main topical in the article. Then, the authors should be revised the proposal of the manuscript or included more references and details about these properties.

A conclusion contain a future perspective should be included in the final of discussion.

Author Response

Dear Reviewer,
Thank you for your comments and suggestions. We revised the article and corrected it.

English is corrected in all manuscript.
1) Abstract is rewritten according to your suggestions. (Line 13-31)
2) All tables are self-made and references are shown. All figures are created using Biorender.com.

Table 1. – Line 76

Table 2. – Line 105

Table 3. – Line 538

Table 4. – Line 593

Table 5. – Line 832

Figure 1. – Line 137

Figure 2. – Line 483

Figure 3. – Line 569

Figure 4. – Line 670

3) Changes are done and references are shown: (Line 128-130)

PRF use is described in alveolar ridge preservation [24–32], orthognathic surgery [33,34], cleft lip and palate surgery [35,36], maxillary sinus augmentation [37–44], dental implant placement [24,45–49].

4) The paragraphs are joined together: (Line 130-136)

It showed the ability to accelerate the closure of an extraction socket, improve the periodontal status, reduce postoperative pain sensation and swelling, accelerate neurosensory recovery, reduce bone resorption, increase and accelerate bone formation, enhance bone stability after surgery and provide faster dental implant osseointegration as well as increased dental implant stability (Figure 1). The majority of studies confirm the clinical advantages of PRF, thus a deeper understanding of its use in several applications is a must. 

5) Changes are done and references are shown: (Line 242-243)

Despite that PRF is rarely described as a biomaterial for use in orthognathic surgery, in some research it demonstrated a favourable effect [33, 34].

6) The paragraphs are joined together. (Line 304-307)

A sinus lift is also a frequent surgical procedure of the atrophic maxilla to augment the maxillary sinus. PRF is widely used for clinical outcome improvement purposes. Some authors described the PRF effect using it alone as a grafting material for maxillary sinus floor augmentation.

7) We included a graphical scheme with the latest research in the last 10 years from databases – Web of Science, PubMed, Scopus, Web of science. (Line 483)

8) The table with information about used bacteria in the studies is included in the article. (Line 538)

9) References are added to the paragraph. (Line 547-559)

As PRF is not characterized as a homogenous fluid but has an extra-cellular matrix with specific architecture and clotting ability, may create difficulties in antimicrobial activity assessment using usual methods. Antimicrobial properties of PRF have been noticed but still are open for discussion because the mechanisms responsible for such property are still unclear [67]. Studies on PRF properties demonstrate heterogeneous results, but the last decade’s studies on PRF antimicrobial activity prove its antimicrobial properties. L-PRF membrane and exudate [17,66,67], I-PRF [60], PRF obtained by horizontal centrifugation (H-PRF) clot, membrane and exudate [16] and not specified types of PRF [65,69,71] were used to describe their antimicrobial properties. The use of the appropriate methodology for antimicrobial activity testing is crucial and should be adjusted in order to avoid false results that could be the reason for controversial assessment of PRF properties. Therefore, in this review, antimicrobial studies done in the last decade were briefly described and analyzed according to the used methodology.

10) This statement is a hypothesis from the authors made after a literature search and analysis regarding the low frequency of the disc diffusion method used in the studies. (Line 573)

The disc diffusion method is rarely described in PRF antimicrobial testing, probably because of the PRF’s ability to make a clot.

11) Changes are done by describing another method of PRF effect on biofilm formation: (Line 717-724)

The specific method of evaluation of PRF activity against biofilm is described by Sanchez et al. In this study, L-PRF exudate effect against P. gingivalis was tested in multi-species pre-formed biofilm or in biofilm during its formation. 14 species community were preformed using BIOSTAT B TWIN bioreactor (Sartorius, Germany). L-PRF did not show a statistically significant decrease of P. gingivalis concentration in pre-formed multi-species biofilm but in developing multi-species biofilm it was able to decrease P. gingivalis concentration in comparison to PBS (p = 0.006), inactive L-PRF exudate and L-PRF exposed to horseradish peroxidase [76].

12) The table with information about antibacterial effect of periodontal pathogens is involved: (Line 593)

13) References are added to the statement. (Line 726-727)

Another very important and at the same time basic property of PRF is its ability to release various growth factors and cytokines to stimulate tissue regeneration [66].

14) The first paragraph of 4.2. section is separated into smaller paragraphs. (Line 765-822)

15) References are added to the paragraph. (Line 833-837)

Scanning electron microscopic evaluation of L-PRF membranes prepared using the Intra-Spin centrifuge showed the observed cells with a normal shape and being alive, besides lymphocytes that presented a textured surface indicating activated cells [81].

16) In the discussion paragraph about PRF’s anti-inflammatory properties is included. (Line 940-946)

PRF effect on cells was shown to be not only stimulating by promoting proliferation and differentiation of osteoblast cells [77] but also was able to suppress the secretion of pro-inflammatory cytokines [79]. PRF demonstrates slow, continuous, and stable release of growth factors compared to PRP [81,82]. Different PRF types showed differences not only in cytokine expression pattern [80], angiogenetic activity [78], and ability to release BMP2 but are working through different signaling pathways [80]. Thereby emphasizing the importance of different types of PRF.

17) The abstract is rewritten emphasizing all 3 sections – clinical application of PRF in oral and craniomaxillofacial surgery, anti-microbial properties, and anti-inflammatory properties.

(Line 13-31)

18) The paragraph about future perspectives is included: (Line 956-961)

PRF is material with great perspectives to be used in further studies to find out and explain properly mechanisms that are involved in its anti-microbial and anti-inflammatory effects. Challenging research direction is such a patient-specific material standardization for clinical outcome prediction. Furthermore, the material has the potential to be used and studied as a drug carrier or drug delivery system and to be incorporated into porous materials for bone tissue regeneration.

Thank you for your collaboration. Look forward to hearing from you.

Kind regards,

Dr. Lana Micko

Reviewer 2 Report

In this manuscript, insight in a wide range of clinical applications of PRF in oral and maxillo-facial surgery and analysis of PRF antimicrobial and anti-inflammatory properties from in vitro studies is provided. The potential of PRF use in medicine by describing its basic antimicrobial and immunological properties as well as to demonstrate the reasons of controversial evaluation of PRF effects are emphasized. The manuscript is well written, though the authors need to address some concerns before it is considered for publication.

1. Delete extra “platelet concentrates” since you firstly used it in Line 55. Please just use the abbreviation “PCs” in the rest of the whole manuscript. Same thing for “PRF” and other abbreviations.

2. When you have numbers and units, please keep them in the format: number [space] unit. For example, Line 316, “408g” should be “408 g”. Please correct the rest of mistakes.

3. I like the figures in this manuscript. They look neat!

Author Response

Dear Reviewer,

We are very grateful for your suggestions. 

1) The corrections regarding terms and abbreviations are done.
2) Spacing mistakes are corrected in the whole manuscript.

Thank you for the collaboration! I look forward to hearing from you.
Kind regards,

Dr. Lana Micko
